# *Phytophthora pseudocryptogea*, *P. nicotianae* and *P. multivora* Associated to *Cycas revoluta*: First Report Worldwide

**DOI:** 10.3390/plants12051197

**Published:** 2023-03-06

**Authors:** Francesco Aloi, Rossana Parlascino, Sebastiano Conti Taguali, Roberto Faedda, Antonella Pane, Santa Olga Cacciola

**Affiliations:** Department of Agriculture, Food and Environment, University of Catania, 95123 Catania, Italy

**Keywords:** *Phytophthora* stem and root rot, sago palm, dieback, leaf baiting, pathogenicity, new disease

## Abstract

A dieback was observed on three-year-old pot-grown plants of *Cycas revoluta* in Sicily (Italy). Symptoms, including stunting, yellowing and blight of the leaf crown, root rot and internal browning and decay of the basal stem, closely resembled the Phytophthora root and crown rot syndrome, common in other ornamentals. Isolations from rotten stem and roots, using a selective medium, and from rhizosphere soil of symptomatic plants, using leaf baiting, yielded three *Phytophthora* species, *P. multivora*, *P. nicotianae* and *P. pseudocryptogea*, were obtained. Isolates were identified by both morphological characters and DNA barcoding analysis, using three gene regions: ITS, *β-tub* and COI. *Phytophthora pseudocryptogea* was the sole species isolated directly from the stem and roots. The pathogenicity of the isolates of the three *Phytophthora* species was tested on one-year-old potted plants of *C. revoluta*, using both stem inoculation by wounding, and root inoculation through infested soil. *Phytophthora pseudocryptogea* was the most virulent and, like *P. nicotianae*, reproduced all the symptoms of natural infections, while *P. multivora* was the least virulent and induced solely very mild symptoms. *Phytophthora pseudocryptogea* was identified as the causal agent of the decline of *C. revoluta*, as it was re-isolated from both the roots and stems of artificially infected symptomatic plants, thus fulfilling Koch’s postulates.

## 1. Introduction

Cycads (order *Cycadales*) are slow-growing, long-living, perennial, dioecious plants with a palm-like habit, robust stems and large evergreen leathery pinnate leaves. Geographically, cycads, in the wild, are restricted to tropical and subtropical or warm, temperate regions of the world, with predominantly summer rainfalls. This group of gymnosperms is often cited as one of the most ancient lineages of living seed plants with an evolutionary history dating back to more than 300 million years ago [1,2,3]. In Asia, cycads are grown in gardens since ancient times and after their first discovery by Europeans in the 18th century, have become highly appreciated as ornamentals worldwide. Many of them are regarded as a threatened species, mainly because they have been massively removed from their natural range for private collections or botanical gardens, landscaping or other commercial purposes, such as pharmaceutical and food uses [1,4]. The living cycads can be divided into three families: *Cycadaceae*, *Stangeriaceae*, and *Zamiaceae*. The current World List of Cycads, which is periodically updated to reflect changes in cycad nomenclature and taxonomy, comprises a total of 10 accepted genera and 370 accepted species [5]. The number of recognized species within each genus is as follows: *Bowenia* (2), *Ceratozamia* (32), *Cycas* (118), *Dioon* (16), *Encephalartos* (65), *Lepidozamia* (2), *Macrozamia* (41), *Microcycas* (1), *Stangeria* (1) and *Zamia* (81). These taxa are distributed throughout the warmer areas of North and South America, Africa, Asia, and Australia, as well as a large number of oceanic islands. Centers of cycad diversity occur in southern Africa, Australia, and the tropical New World. All three cycad families are represented in Australia and Africa, whereas only *Cycadaceae* and *Zamiaceae* are endemic in Asia and the New World, respectively. At the generic level, the New World and Australia show the greatest diversity, each with four genera (*Ceratozamia*, *Dioon*, *Microcycas*, and *Zamia* in New World; *Bowenia*, *Cycas*, *Macrozamia* and *Lepidozamia* in Australia). Three genera are endemic in Africa (*Cycas*, *Encephalartos*, and *Stangeria*) and only *Cycas* occurs in Asia [1]. *Cycas* is the genus type and the only genus currently recognized in the family *Cycadaceae*. Diversification analyses based on time-calibrated phylogenetic trees point at Indocina as the ancestral geographic area of the origin of *Cycas*, which evolved more recently than other members of *Cycadales* [3,6]. The most popular species of this genus is *Cycas revoluta*, known with diverse names, the most common of which is the Sago Palm or King Sago Palm, referring to both its palm-like appearance and sago, a starch extract used as a traditional food by populations of Southeast Asia. This cycad is believed to be native to Japan and southern China, but is cultivated as an ornamental and landscape plant in warm and temperate climates all over the world. It is common in botanical and private gardens and is easily propagated by seeds or offshoots.

In April 2010, for the first time, 30% of a production block of approximately 8000 three-year-old potted sago palm plants, grown in open air in a commercial nursery of ornamentals in the Catania province (Eastern Sicily, Italy), showed severe symptoms of dieback. Overall, symptoms closely resembled the Phytophthora root rot (PRR) syndrome, which is common in nurseries of ornamentals worldwide [7,8,9,10,11,12,13,14,15,16,17,18]. For instance, *Phytophthora nicotianae* and *P. palmivora* were responsible for a similar disease of potted ornamental palms reported a few years ago in Sicily [19,20]. Moreover, *P. cinnamomi* was identified as the causal agent of a severe dieback associated with stem and root rot of Modjiadi cycad (*Encephalartos transvenosus*), within the native range of this cycad in South Africa [21]. The same *Phytophthora* species was reported as a pathogen of other cycads, including species of *Cycas* in New Zealand, *Macrozamia* in Australia and *Zamia* in Taiwan [21], and Johnson’s cycad (*Macrozamia johnsonii*) is in the list of native plant species threatened by *P. cinnamomi* in New South Wales (Australia) [22]. Since the first record in 2010, the disease has been under observation and has occurred sporadically and with a low incidence in several nurseries in the same production area of Sicily. However, in May 2021, it appeared again in a severe form and with an incidence comparable with its first record, in a stock of 15,000 three-year-old potted plants of sago palm grown in open air, in a nursery where the disease had not been noticed before. The severity of this new disease outbreak prompted us to investigate a possible involvement of *Phytophthora* in its etiology.

## 2. Results

### 2.1. Symptoms of the Disease

Symptoms consisted of stunting, chlorosis and blight of the leaflets or entire leaves, initially affecting only part of the leaf crown and, subsequently, expanding to the other leaves of the crown and resulting in the withering of the entire plant in most severe cases. Foliar symptoms were associated with the internal browning and the decay of the stem base and root rot (Figure 1).

### 2.2. Morphological Identification of Isolates

In agreement with a provisional diagnosis based on the symptoms, isolates of a *Phytophthora* sp. were consistently obtained from rotten tissues of diseased sago palm plants from the nursery in the province of Catania, where the disease was first noticed and the isolates were preserved as living cultures in the collection of the Molecular Plant Pathology Laboratory of the Department of Agriculture, Food and Environment of the University of Catania, Catania, Italy. Since then, the disease has been under observation and has occurred sporadically and with a low incidence in several nurseries. However, in May 2021, it appeared again in a severe form and with an incidence comparable to its first record, in a stock of 15,000 three-year-old potted plants of sago palm grown in open air, in a nursery where the disease had not been noticed before.

Direct isolation from rotten roots and stems of sago palm plants consistently produced only one species of *Phytophthora*, as tentatively determined based on the morphology of colonies, with a mean isolation frequency of 75%. Overall, representative single-hypha isolates (isolates from roots and stems) sourced in 2021 were selected and characterized together with isolates obtained from roots and stems of symptomatic plants in 2010, which shared the same culture morphology (Table 1).

All the isolates formed colonies with chrysanthemum pattern on V8A and grew between 3 °C and 35 °C on CMA, with the optimum at 25 °C. Sporangia were non-papillate, persistent, terminal, ovoid and ellipsoid to obpyriform (23–38 × 35–62 µm), originating in unbranched or simple sympodial sporangiophores (Figure 2 and Figure 3). Hyphal swellings were globose to irregular and produced in visible networks. Chlamydospores were absent. In dual cultures, all isolates produced gametangia and oospores only when mated with the A2 tester strain of *P. cryptogea*, indicating they were all of A1 mating type. Oogonia were globose with a smooth wall, some with tapered bases (diameter ranging from 27.3 to 37.0 μm). Antheridia were amphigynous, and oospores aplerotic to nearly plerotic (diameter ranging from 24 to 32 µm), with thick walls (2.5 µm).

By contrast, three different species of *Phytophthora* were recovered by leaf-baiting from rhizosphere soil of symptomatic plants. Two species were heterothallic and did not form gametangia in the single culture, whereas the third species was homothallic and produced gametangia and oospores in the single culture. The two heterothallic species were recovered from the soil of 20 and 18 pots, respectively, of the 20 sampled pots, while the homothallic species was recovered from 11 out of 20 pots. Overall, the proportions of isolates of each species recovered from potting soil were 48%, 33% and 19%, respectively.

The first heterothallic species, the dominant one and the only one recovered from all pots, showed the same colony morphology of the species recovered from the roots and stems. All isolates were of A1 mating type. The second heterothallic species was distinguishable from the previous one as it formed coralloid colonies on V8A and grew between 9 °C and 33 °C on CMA, with the optimum at 28 °C (Figure 2). Sporangia were persistent, papillate, spherical to ovoid, ellipsoid and obpyriform (24–63 × 15–46 µm) (Figure 3). Hyphal swellings were globose, sub-globose and intercalary, with radiating hyphae. The majority (66%) of the isolates of this species were A2 mating type, while 34% were of A1 mating type. Oogonia were spherical with smooth walls (mean diameter 26 ± 2 µm), amphigynous antheridia and aplerotic oospores (20–27 µm diam.). Spherical, intercalary chlamydospores were produced. The third species was homothallic and produced gametangia and oospores in the single cultures. It formed stellate colonies on V8A and grew on CMA at the optimum temperature of 25 °C, minimum 5 °C and maximum 32.5 °C (Figure 2). Sporangia were semi-papillate, often with two or three papillae, persistent, and ovoid, limoniform, or ellipsoid (37–56 × 13–31 µm) (Figure 3). Hyphal swellings and chlamydospores were absent. Oogonia were globose, with smooth walls (mean diameter 26.3 ± 1.7 µm), oospores were nearly plerotic, globose with thick walls (17–30 µm diam.), and antheridia paragynous.

### 2.3. Molecular Identification

ITS, *β-tub* and COI sequences of all isolates obtained from the stems and roots, and from the prevailing species among the three *Phytophthora* species recovered from rhizosphere soil of *C. revoluta,* and separated preliminarily based on the colony morphology, revealed 100% identity with the corresponding sequences of *P. pseudocryptogea* ex-type isolate CBS139749 = VHS16118 (Accession Nos. KP288376: ITS, KP288392: β-tubulin, KP288342: COI) and other reference isolates of the same species. Interestingly, the isolates obtained from symptomatic *C. revoluta* plants sampled in 2010, when the disease was noticed for the first time, belonged to this species. The isolates of the other two species of *Phytophthora* recovered from rhizosphere soil showed 99/100% sequence identity for all three barcoding genes with those of well-authenticated specimens of *P. nicotianae* (e.g., Accession Nos. MG865550: ITS, MH493986: β-tubulin, MH136943: COI) and the ex-type CBS 124094 and other reference isolates of *P. multivora* (Accession Nos. FJ237521: ITS, FJ665260: β-tubulin, MH136939: COI), respectively. Since the isolates belonging to the same species were all identical, only three isolates, one per species, were used for the pathogenicity assays.

The sequences generated in this study have been deposited in the NCBI-GenBank and accession numbers are given in Table 1.

### 2.4. Pathogenicity Tests

Both the progression and severity of symptoms did not differ significantly between plants inoculated on the stem by wounding and plants inoculated on roots through infested soil.

The first symptoms consisting of leaf yellowing were observed on plants inoculated with the isolate CIPH2 of *P. pseudocryptogea,* six days post-inoculation (dpi). Later, at 24 dpi, the chlorotic leaves withered and the value of the disease severity index (DSI) ranged between 2 and 3. Sixty dpi in plants inoculated with *P. pseudocryptogea*, symptoms further evolved and the DSI values ranged between 3 and 4.

On plants inoculated with the isolate CycC11R of *P. nicotianae*, the leaves turned yellow at 24 dpi (DSI ranging from 1 to 2) and were withered at 36 dpi (mean DSI 2). Conversely, the plants inoculated with the isolate CycC2 of *P. multivora* showed only mild leaf symptoms and the control plants remained completely asymptomatic for the duration of the test (Figure 4).

All plants of *C. revoluta*, tested both by wound-inoculation and soil-infestation with the isolates CIPH2 of *P. pseudocryptogea* and CycC11R of *P. nicotianae*, developed symptoms of leaf chlorosis, root and stem rot within two months after the transplanting, (Figure 5), although symptoms were more severe on plants inoculated with *P. pseudocryptogea*. Conversely, plants inoculated with the isolate CycC2 of *P. multivora* and the control plants did not show symptoms of stem and root rot.

The stem damage class for control plants was zero in both experiments (soil-infestation and wound-inoculation) and the mean value of fine root/main root weight (frw/mrw) ratio in soil-infestation and stem wound-inoculation tests were 0.87 ± 0.12 and 0.94 ± 0.21, respectively (Figure 6, Figure 7 and Figure 8). *Phytophthora pseudocryptogea* (CIPH2) was the most aggressive species, causing withering of 70% of sago palm plants within two months after the transplanting. It had a frw/mrw ratio of 0.42/0.34 (soil-infestation/wound-inoculation method), (51.7% and 63.8% reduction compared to the control) and a stem damage class of 3.5 (i.e., 76.5% extensive stem dieback) in the soil-infestation test and 3 (i.e., 65% advanced stem dieback with large necrotic lesions) in the wound-inoculation test (Figure 6, Figure 7 and Figure 8). *Phytophthora nicotianae* (CycC11R) caused withering of 30% of the plants, a frw/mrw ratio of 0.42/0.50 (51.7% and 42.5% reduction compared to the control), and a stem damage class of 1.5 in the soil-infestation test and 1 in the wound-inoculation test (Figure 6, Figure 7 and Figure 8). Differences in the stem damage class and frw/mrw ratio compared to the control were statistically significant for both isolates CIPH2 (*P. pseudocryptogea*) (stem damage class: *p* < 0.01 in both test; frw/mrw ratio: *p* < 0.001 in the soil-infestation test, *p* < 0.01 in the wound-inoculation test) and CycC11R (*P. nicotianae*) (stem damage class: *p* < 0.001 in the soil-infestation test, *p* < 0.05 in the wound-inoculation test). In plants inoculated with *P. multivora* (CycC2), the stem damage class and frw/mrw ratio values were not statistically significant compared to the control.

*Phytophthora pseudocryptogea* was re-isolated consistently, from both symptomatic roots and stems, thus fulfilling Koch’s postulates. *Phytophthora nicotianae* was re-isolated consistently only from the roots and only very sporadically from the stems. *Phytophthora multivora* was re-isolated in a very low proportion and solely from the roots. Each species of *Phytophthora* was re-isolated from the infested potting soil used in root-inoculation tests. The identity of isolates obtained from necrotic roots and stem rot of symptomatic, artificially inoculated sago palm plants, was determined by the colony morphology, microscopic observations and rDNA ITS sequencing.

In summary, *P. pseudocryptogea* proved to be pathogenic to *C. revoluta* plants, causing stem and root rot, and was the most virulent among the three species tested. Additionally, *P. nicotiane* caused stem and root rot, but was less virulent than *P. pseudocryptogea*. *Phytophthora multivora* resulted in being only weakly pathogenic to this host and induced mild symptoms on the leaf crown, while no significant effect of inoculation was evident on the stem and roots compared to the non-inoculated control.

## 3. Discussion

In this study, three diverse *Phytophthora* species, *P. multivora* (ITS clade 2c), *P. nicotianae* (ITS clade 1) and *P. pseudocryptogea* (ITS clade 8a)*,* were found to be associated with the dieback syndrome of pot-grown plants of *C. revoluta* observed in Sicily. Of the three species, only *P. pseudocryptogea* was isolated from infected tissues of stems and roots of *C. revoluta* plants with natural infections, while *P. multivora* and *P. nicotianae* were recovered solely from the rhizosphere soil. Moreover, *P. pseudocryptogea* was isolated from symptomatic plants when the disease was first recorded, and again 11 years later when it reappeared in a distinct nursery with the same incidence and severity as before. This was also the *Phytophthora* species recovered consistently and most frequently from the rhizosphere soil of symptomatic plants. In pathogenicity tests, *P. pseudocryptogea* was the most virulent among the three species recovered from potting soil of sago palm plants and induced typical symptoms of the disease on leaves, stems and roots. Finally, it was re-isolated consistently from symptomatic, artificially infected plants, thus fulfilling Koch’s postulates. Based on this experimental evidence, *P. pseudocryptogea* was identified as the pathogen responsible for the dieback syndrome of *C. revoluta* occurring in Sicilian nurseries. *Phytophthora pseudocryptogea* has been segregated as a separate species from the *P. cryptogea* species complex, based prevalently on multiple gene phylogenetic analyses and described formally in 2015 as a new species in clade 8a distinct from *P. cryptogea sensu stricto* [23]. Previous studies demonstrated that *P. pseudocryptogea* isolates from diverse continents and hosts are genetically uniform [23,24]. It was hypothesized that this homogeneity may be due to the global distribution of a single clone and the lack of sexual reproduction since, consistently with the present study, all isolates of this species so far characterized are of the A1 mating type [23]. In addition, *P. pseudocryptogea* isolates from *C. revoluta* characterized in this study and those examined by Safaiefarahani et al. [23], who first described *P. pseudocryptogea* as a distinct species, share identical double bases at the same position in ITS sequences, strengthening the hypothesis that they belong to the same clonal population and confirming that *P. pseudocryptogea* is an introduced pathogen in Sicily. Until recently, the geographic distribution of *P. pseudocryptogea* was believed to be restricted to Asia (Iran), Australia and South America (Ecuador) [25]. However, in the last years, this species has been repeatedly recovered from nurseries and forest stands in the two major Italian islands (Sardinia and Sicily), Spain and Turkey, indicating it has established and is widespread in the Mediterranean region [18,26,27,28,29,30,31]. The known host range of this *Phytophthora* species is being rapidly expanding and presently includes species of the *Aquifoliaceae*, *Asparagaceae, Asphodeliaceae*, *Cupressaceae*, *Fagaceae, Lauraceae, Oleaceae, Platanaceae, Proteaceae, Salicaceae* and *Solanaceae* families [18,25,26,27,28,29,30,31]. Therefore, this is not the first report of *P. pseudocryptogea* on a gymnosperm, but the first one as a pathogen of a cycad worldwide. Possibly, reports of *P. cryptogea* on other ornamentals in Sicily, prior to the introduction of multiple gene phylogenetic analyses as a taxonomic and diagnostic criterion and the definition of *P. pseudocryptogea* as a distinct species within the *P. cryptogea* complex, have to be critically reconsidered [32,33,34]. This study confirms that *P. pseudocryptogea* has entered the nursery supply chain in Italy, and like other *Phytophthora* species introduced recently, such as *P. niederhauserii* [12,35,36], it may be regarded as an emerging pathogen in the Italian nursery industry [37].

*Phytophthora nicotianae*, the second species recovered most frequently from rhizosphere soil of symptomatic *C. revoluta* plants in this study, is a cosmopolitan and very polyphagous pathogen, as well as one of the most prevalent *Phytophthora* species in ornamental nurseries in Italy [13,37,38,39,40,41,42,43,44]. Like other soilborne *Phytophthora* species, even though it is prevalently a root pathogen, in ornamental nurseries under favorable environmental conditions, it may adapt temporarily to an aerial life style, causing aboveground infections [45,46]. Differently from *P. pseudocryptogea*, both A1 and A2 mating types of *P. nicotianae* were recovered from the potting soil of *C. revoluta* plants, in agreement with previous reports, indicating that both mating types of this species occur in nurseries of ornamentals in Italy [39,40,47]. The presence of both mating types in the same nursery is a prerequisite for this heterothallic species to reproduce sexually and the demonstration of its ability to give rise spontaneously to interspecific hybrids confirm sexual reproduction is not a rare event in the nursery of ornamentals [48,49]. Genetic recombination through sexual reproduction might favor the emergence of new genotypes and the expansion of the already broad host range of this pathogen [50]. Moreover, the ability to form resting structures, such as oospores or chlamydospores, may condition other aspects of the biology of *P. nicotianae*, such as the tolerance of extreme environmental conditions. The ecological plasticity of this species and its polyphagy may explain why it is so common in the nursery industry of southern Italy. In pathogenicity tests, *P. nicotianae* induced decay symptoms in both stems and roots of artificially inoculated plants of *C. revoluta*, but was relatively more virulent to roots than stems, indicating sago palm as a potential host this *Phytophthora* species, thus expanding the list of its hosts, and confirming it is prevalently a root pathogen [44,51]. 

*Phytophthora multivora* was formally described as a separate species of the *P. citricola* species complex in 2009 [52]. Later, it was found to be common in natural ecosystems in South Africa [53]. However, there it did not cause any severe disease outbreaks, suggesting a long-term co-evolution with the native tree species. Presently, *P. multivora* has a wide geographical distribution, encompassing Western Australia, South Africa, New Zealand, USA, Europe and the Canary Islands, probably as a consequence of the global trade of living plants and plant material [53]. The genetic analysis of a large collection of isolates from all over the world, using two diverse sets of molecular markers, pointed at South Africa as the center of origin of this *Phytophthora* species [53]. In Sicily, *P. multivora* is common in protected natural areas, vegetation conservation sites and ornamental nurseries [26,29,30,54]. Although this *Phytophthora* species is regarded an emerging pathogen worldwide and has a broad host range, also including *Macrozamia riediei*, a cycad native to Australia [25], in pathogenicity tests on *C. revoluta,* it was weakly pathogenic, causing only mild symptoms in leaves and no visible decay of the stem and roots, confirming it was not the causal agent of the dieback syndrome of this cycad recorded in Sicilian nurseries. The presence of recently introduced exotic species, such as *P. multivora* and *P. pseudocryptogea*, or a species preferentially associated with agricultural systems, such as *P. nicotianae*, in natural ecosystems and nurseries in southern Italy, would confirm the role of the nursery industry in the introduction and spread of invasive *Phytophthora* species, as evidenced for *P. ramorum* [55]. All three *Phytophthora* species recovered from potting soil in this study are polyphagous and the nursery where the disease was detected was not specialized in the production of sage palm. Consequently, any hypothesis concerning the introduction and spread of these pathogens in the nurseries would be speculative. 

In general, the contemporary presence of diverse *Phytophthora* species, including species known as aggressive pathogens, is common in ornamental nurseries. However, in many cases, not all the species cause disease and some species, despite their polyphagia, privilege certain hosts, although evidence of host specialization has been provided only in a few cases. The association of multiple *Phytophthora* species with a specific host inspired the concept that in nurseries of ornamental plants, PRR has to be regarded as a complex disease [56]. However, potentially pathogenic *Phytophthora* species may be recovered from soil of asymptomatic plants and not all *Phytophthora* species detected in the soil of symptomatic plants are causing disease. To make PRR management strategies in nurseries more effective, it would be useful to complement the analysis of the *Phytophthora* species community associated with the nursery industry, with the study of the lifestyle of these species.

## 4. Materials and Methods

### 4.1. Isolation and Morphological Characterization of Isolates

Isolations were made from 20 symptomatic sago palm (*C. revoluta*) plants with different disease severities, collected in 2021 from a single nursery in the Catania province, Sicily (Southern Italy). Plants were uprooted, washed with tap water and cut off. Rotten roots and stems were washed again with running tap water, immersed in 1% NaClO for 2 min and then in 70% EtOH for 30 s, rinsed in sterile distilled water, and dipped dry before being cut into 5 mm pieces. Subterranean stem and root tissue pieces were plated onto selective BNPRAH V8 agar medium [57] in Petri dishes (five pieces per dish), and then incubated for 24–72 h in the dark at 25 °C. Pure cultures of the same colony pattern were obtained by subculturing single hyphae onto V8 juice agar (V8A) [58]. Potting-soil samples were collected from the same 20 pots with symptomatic plants used for direct isolations from tissues. Subsamples of ca. 200 mL soil per pot were used for the leaf baiting test, which was performed in a walk-in growth chamber with 12 h natural day light at 20 °C, according to the protocol described by Riolo et al. [59]. Young leaves of native plant species (*Ceratonia siliqua* and *Quercus* spp.) were used as baits. Necrotic segments (2 × 2 mm) from infected leaves were plated onto selective PARPNH agar [60]. Petri dishes were incubated at 20 °C in the dark. Outgrowing *Phytophthora* hyphae were transferred onto V8A under the stereomicroscope. All isolates were maintained on the V8A and stored at 6 °C in the dark. Isolates were in-depth characterized based on the colony morphology. Isolates obtained from symptomatic sago palm in 2010 and preserved in the collection of the Molecular Plant Pathology Laboratory, at the University of Catania, were also included in this study. All *Phytophthora* isolates characterized in this study are listed in Table 1. However, since all isolates of the same species showed the same features, only a few of them are listed in the table.

Morphological features of isolates, including the morphology and dimensions of reproductive structures, and colony shape, were determined on V8A and PDA at 25 ± 1 °C in the dark, according to standard procedures [58]. Sporangia production was stimulated following the method described by Jung et al. [61]. Small fragments (size 2 mm) were cut from the growing edge of 5- to 7-day-old cultures grown in Petri dishes (15 mm diam) on V8A at 20 °C in the dark. They were then placed in a 5 cm diameter Petri dish and flooded with non-sterile soil extract water (200 g soil suspended in 1 L of de-ionized water for 24 h at room temperature and then filtered). After incubation at 20 °C in the dark for 24–72 h, dimensions and morphological features of 50 mature sporangia of each isolate were determined at ×400 magnification. For cardinal temperatures, the isolates were grown on CMA (Corn Meal Agar; Oxoid Limited, Basingstoke, UK) dishes and incubated at 1, 3, 5, 10, 15, 20, 25, 30, 35 and 37 °C, and radial colony growth was measured after 2–10 days. There were three replicate plates for each isolate and temperature. 

The sexual compatibility type of *P. pseudocryptogea* isolates from cycas was determined by crossing *Phytophthora* isolates with A1 and A2 mating types of *P. cryptogea* (CBS 113.119) and *P. drechsleri* (CBS 292.35), respectively. Mycelial plugs (5 mm diameter) of the unknown mating type isolate and either CBS 113.119 or CBS 292.35 isolates were placed 4 cm apart from each other on 10% clarified V8A medium. The plates were incubated for 10 days at 20 °C in the dark or until oospores were produced. The mating type of *P. nicotianae* isolates from cycas was determined using the same method, and the *P. nicotianae* isolates IMI 268688 (A1) and Albicocco9 (=IMI 396203) (A2), as testers [62,63]. All the mating type tests were repeated in duplicate.

### 4.2. Molecular Identification

DNA-based identification of 16 representative *Phytophthora* isolates from cycas (Table 1) was performed by sequence analysis of internally transcribed spacer of ribosomal DNA (ITS-rDNA) region, β-tubulin (*β-tub*) and cytochrome c oxidase subunit 1 (COI). DNA was extracted from 7-day-old cultures grown on V8A at 20 °C, by using the PowerPlant Pro DNA isolation Kit (MO BIO Laboratories, Inc., Carlsbad, CA, USA), following the manufacturer’s instructions. ITS-rDNA was amplified using forward primers ITS6 [64] and the reverse primer ITS4 [65]. All PCRs were carried out in a 25 µL reaction mix containing the PCR buffer (1X), dNTP mix (0.2 mM), MgCl_2_ (1.5 mM), forward and reverse primers (0.5 mM each), Taq DNA Polymerase (1 U) and 100 ng of DNA. The thermocycler conditions were as follows: 94 °C for 3 min, followed by 35 cycles of 94 °C for 30 s, 55 °C for 30 s, and 72 °C for 30 s, and then 72 °C for 10 min [64]. β-tubulin and COI were amplified by using the primer pairs TUBUF2/TUBUR1 and COXF4N/COXR4N, respectively [66]. The thermocycle sequence was as follows: an initial denaturation at 94 °C for 2 min, 35 cycles consisting of denaturation at 94 °C for 30 s, annealing for 30 s, and extension at 72 °C for 60 s, and a final extension at 72 °C for 10 min. Annealing temperatures were 60 °C for β-tubulin and 52 °C for COI.

Amplicons were detected in 1% agarose gel and sequenced in both directions by an external service (Macrogen, Amsterdam, The Netherlands). All resultant chromatograms were checked for correct base-calling using FinchTV v.1.4.0. The consensus sequences were first compared by the BLAST program to the gene sequences in the NCBI databases, and then with those of the ex-types and other well-authenticated specimens of *Phytophthora*, using the Clustal 2.1 software.

### 4.3. Pathogenicity Tests

The pathogenicity of *Phytophthora* spp. isolates from sago palm plants was tested in two separate experiments, using the soil-infestation method and the method of stem inoculation by wounding, as described by Aloi et al. [67]. The isolates of *P. pseudocryptogea* (CIPH2), *P. nicotianae* (CycC11R) and *P. multivora* (CycC2) were used to inoculate six-month-old potted plants of sago palm (10 plants for each isolate). Pathogenicity tests by wound inoculation were performed by inserting a mycelial plug of 7-day-old colonies, grown on V8A, into a wound made with a scalpel in the subterranean stem and the tissue patch was sealed with Parafilm^®^. Control plants were inoculated with sterile agar plugs. In soil-infestation pathogenicity assays, cycas plants were transplanted into 12 cm diameter pots containing a mixture of 1:1 steam-sterilized, sandy loam (vol/vol) with 4% inoculum produced on autoclaved wheat kernel seeds. Inoculum consisted of a 21-day-old culture of the isolates grown in the dark at 25 °C in 750 mL Erlenmeyer flasks, containing a sterilized medium made of 50 mL of wheat seeds and 50 mL V8 juice broth. Ten control plants were transplanted in free-draining pots containing non-infested potting mixtures. For both inoculation methods, all plants were maintained in saturated soil for 48 h and then transferred to a growth chamber at 23 °C, 80% relative humidity, and a photoperiod of 16 h of light and 8 h of dark. The disease severity index (DSI) was assessed visually at 6-day intervals, since the appearance of first symptoms as frond yellowing or wilting using a 0–4 rating scale (0 = no symptoms, 1 = 1–33% frond yellowing or wilting, 2 = 34–66% frond yellowing or wilting, 3 = 67–100% frond yellowing or wilting, 4 = dead plant). Symptoms were evaluated at four time intervals after inoculation, i.e., 12, 24, 36, and 60 days post-inoculation (dpi). The trials were considered concluded when the plants showed severe symptoms of withering (60 dpi). Data were analyzed by using a one-way ANOVA, followed by a Tukey ‘s HSD test (honestly significant difference) as a post-hoc test (R software). Differences at *p* ≤ 0.05 were considered significant. At the end of the tests, from all plants showing necrosis of the stem, the severity of stem rot was rated according to a scale of five stem damage classes: 0 = healthy stem; 1 =< 25% slight stem rot; 2 = 26–50% stem rot, beginning stem dieback with small necrotic lesions; and 3 = 51–75% stem rot, advanced stem dieback with large necrotic lesions; 4 = 76–100% stem rot, extensive stem dieback. The roots were dried for 72 h at 65 °C and the dry weights of the main roots (diam. 2–10 mm) and fine roots (diam < 2 mm) were recorded for each plant. Data were analyzed by one-way analysis of variance (ANOVA), followed by a Dunnett’s multiple comparisons test. Statistical analysis was carried out by the R software and considered as significant at *p* < 0.05. Re-isolations from necrotic stems and roots were performed on the BNPRAH V8A medium and the identity of each *Phytophthora* species was confirmed by morphological and molecular analyses.

## 5. Conclusions

Three *Phytophthora* species, *P. multivora*, *P. nicotianae* and *P. pseudocryptogea*, were recovered from the rhizosphere soil of pot-grown sago palm (*Cycas revoluta*) with symptoms of PRR. Of the three *Phytophthora* species recovered from the potting soil of sago palm plants, Koch’s postulates were completed only for *P. pseudocryptogea*, which was identified as the etiological agent of the disease. Conversely, neither of the other two species was isolated from infected tissues of symptomatic plants with natural infections, and in artificial inoculation tests on sago palms, only *P. nicotianae* was proven to be an aggressive pathogen, while *P. multivora* induced very mild symptoms. The role of both *P. multivora* and *P. nicotianae* in PRR of sago palm deserves to be investigated further. It is noteworthy that PRR outbreaks were observed exclusively in plant blocks grown in open air, suggesting the hypothesis that they were triggered by conducive environmental conditions occurring outside the greenhouse, such as soil waterlogging due to heavy rains in winter and aerial dispersal of the inoculum via splashing water.

## Figures and Tables

**Figure 1 plants-12-01197-f001:**
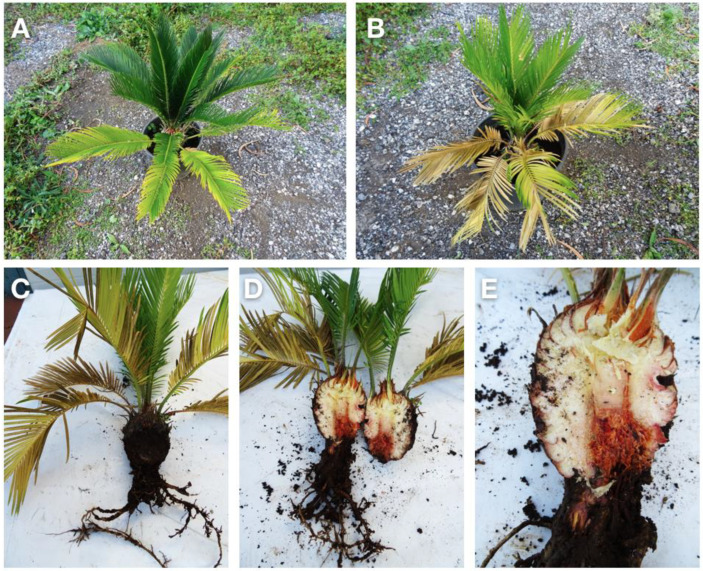
Symptoms of chlorosis, (**A**) followed by desiccation (**B**) of the older leaves and root rot, associated with the internal browning of the subterranean stem on three-year-old potted sago palm plants (**C**–**E**).

**Figure 2 plants-12-01197-f002:**
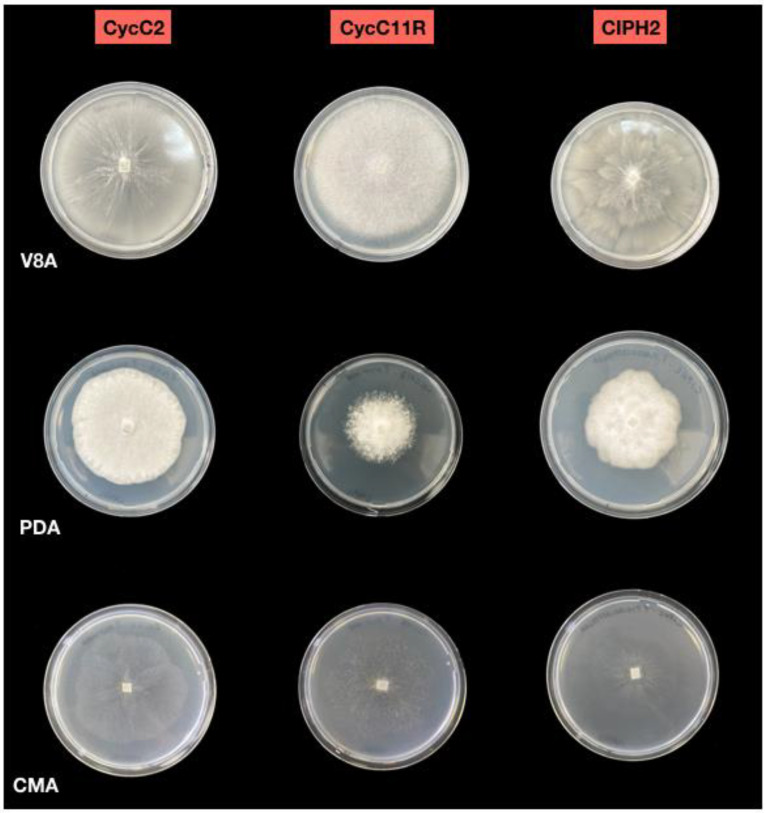
Colony morphologies of *Phytophthora multivora* (CycC2), *P. nicotianae* (CycC11R) and *P. pseudocryptogea* (CIPH2) after 7 days of growth at 20 °C in the dark on V8A, PDA and CMA.

**Figure 3 plants-12-01197-f003:**
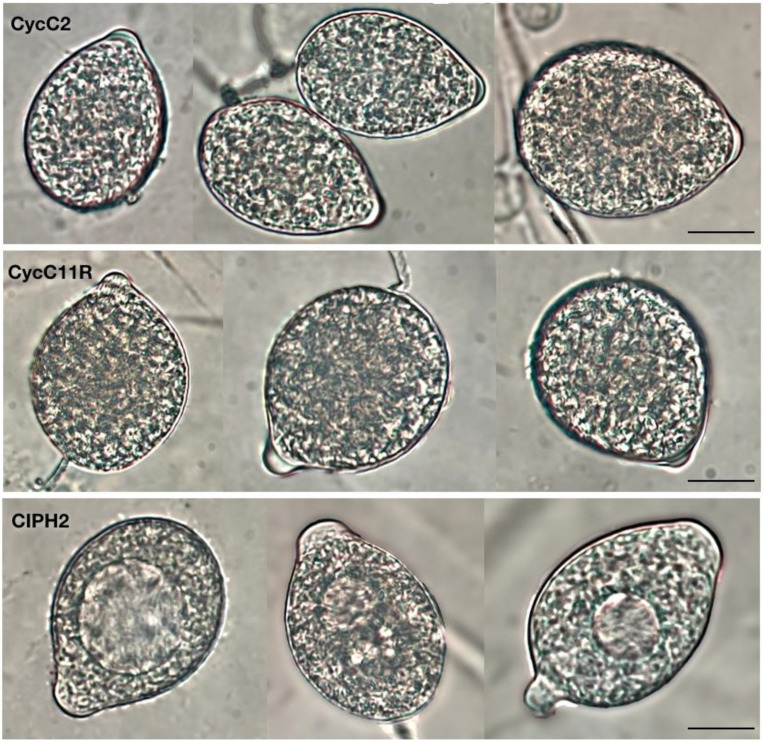
Semi-papillate, persistent, ovoid, limoniform, or ellipsoid sporangia of *Phytophthora multivora* (CycC2). Papillate, persistent, spherical to ovoid, ellipsoid, obpyriform sporangia of *P. nicotianae* (CycC11R). Non-papillate, persistent, terminal, ovoid, ellipsoid to obpyriform sporangia of *P. pseudocryptogea* (CIPH2). Scale bar: 25 μm.

**Figure 4 plants-12-01197-f004:**
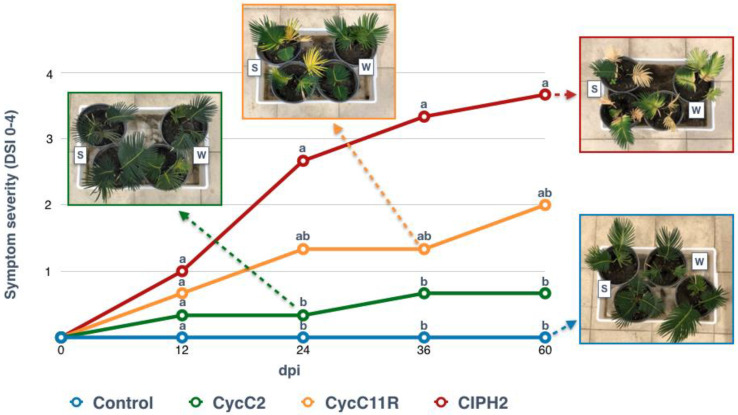
Symptom progression in *Cycas revoluta* plants inoculated (soil-infestation (S) and wound-inoculation method (W)) with *Phytophthora multivora* (CycC2), *P. nicotianae* (CycC11R) and *P. pseudocryptogea* (CIPH2). Non-inoculated plants served as a control. Symptom severity was expressed as the mean value of the disease severity index (DSI) at 12, 24, 36, and 60 days post-inoculation (dpi). Images show the type of symptoms at each time interval post-inoculation. At each time interval, values sharing the same letters are not significantly different according to the Tukey’s honestly significant difference (HSD) test (*p* ≤ 0.05).

**Figure 5 plants-12-01197-f005:**
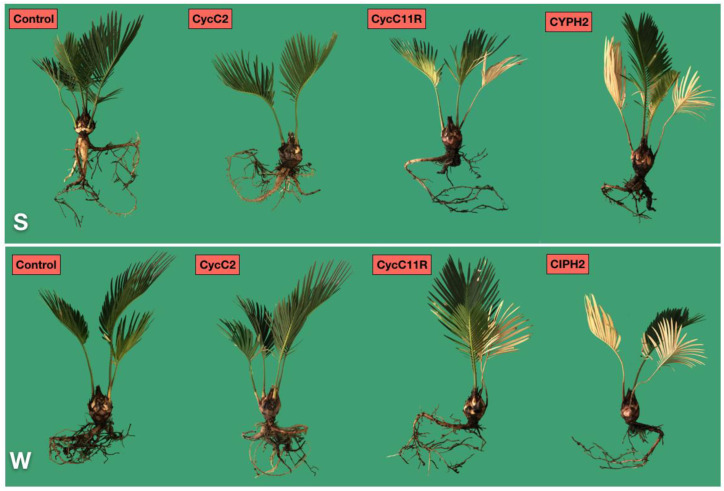
Symptoms of root rot and leaf chlorosis on *Cycas revoluta* plants, 2 months after transplanting from two separate experiments, soil-infestation (S) and wound-inoculation method (W) with *Phytophthora pseudocryptogea* (CIPH2) and *P. nicotianae* (CycC11R); no symptoms appear in the sago palm plants transplanted in both test with *P. multivora* (CycC2). Control plants remained asymptomatic in the two tests.

**Figure 6 plants-12-01197-f006:**
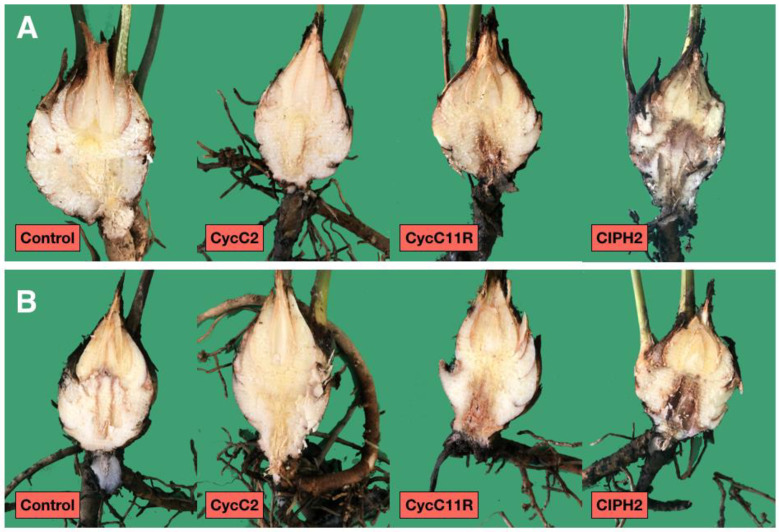
Internal stem necrosis from representative samples of Cycas revoluta plants from two separate experiments, the soil-infestation test (**A**) and the wound-inoculation test (**B**) with Phytophthora pseudocryptogea (CIPH2) and *P. nicotianae* (CycC11R), observed at the end of the two experiments (60 dpi). No symptoms appear in the stems inoculated with *P. multivora* (CycC2) in both tests. Stem from the control plants (control) remained asymptomatic.

**Figure 7 plants-12-01197-f007:**
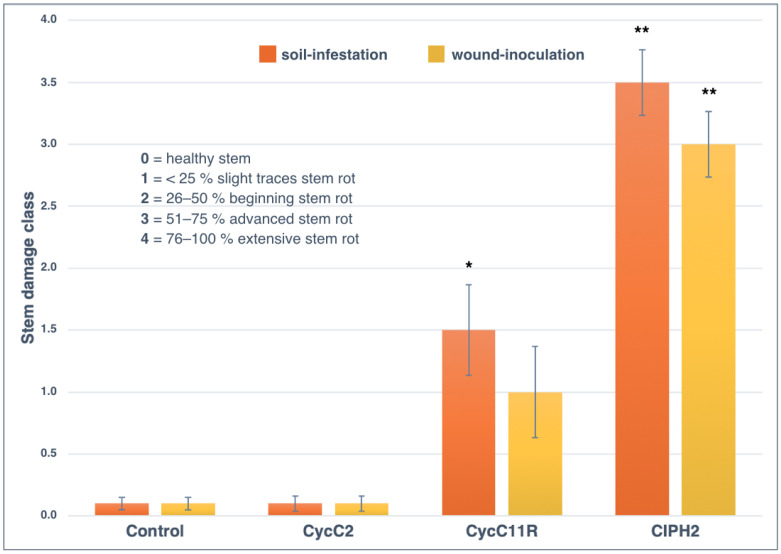
The mean stem damage class of *Cycas revoluta* plants two months after transplanting in the soil-infestation and wound-inoculation tests with non-infested soil/non-inoculated control (control), *P. multivora* (CycC2), *P. nicotianae* (CycC11R) and *P. pseudocryptogea* (CIPH2). Bars show standard deviations; asterisks represent statistical significances (** = *p* < 0.01; * = *p* < 0.05).

**Figure 8 plants-12-01197-f008:**
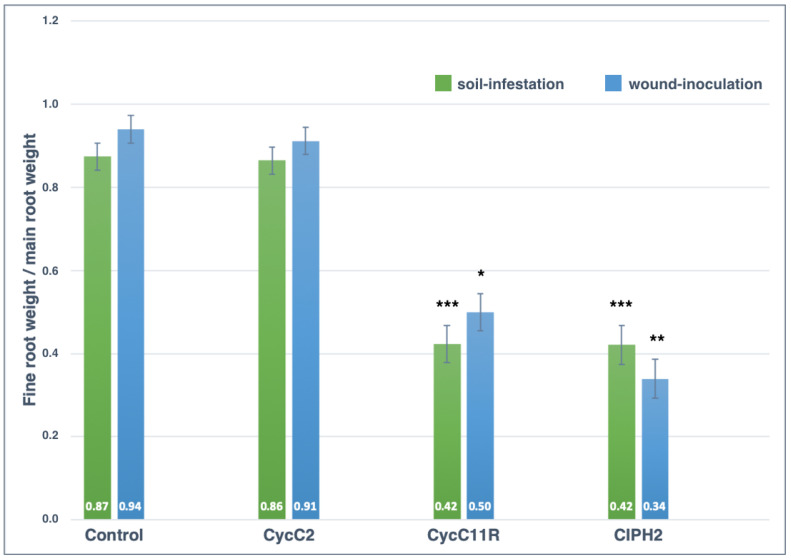
The mean fine root weight/main root weight ratio of *Cycas revoluta* plants 2 months after transplanting in the soil-infestation and wound-inoculation tests with non-infested soil/non-inoculated control (control), *P. multivora* (CycC2), *P. nicotianae* (CycC11R) and *P. pseudocryptogea* (CIPH2). Bars show standard deviations; asterisks represent statistical significances (*** = *p* < 0.001; ** = *p* < 0.01; * = *p* < 0.05).

**Table 1 plants-12-01197-t001:** Isolates of *Phytophthora* from the basal stem, roots and rhizosphere soil of sago palm characterized in this study.

Species	Isolate Code	Source	Mating Type	Year of Isolation	Genbank Accession No.		
					ITS	TUB	COI
*P. pseudocryptogea*	CIPH	Stem	A1	2010	HM627524	OP629924	OP563125
	CIPH2				OP557985	OP629925	OP563126
	CIPH3R	Roots			OP557986	OP629926	OP563127
	CIPH4R				OP557987	OP629927	OP563128
	CycS1	Stem		2021	OP557988	OP629928	OP563129
	CycS2				OP557989	OP629929	OP563130
	CycR1	Roots			OP557990	OP629930	OP563131
	CycR2				OP557991	OP629931	OP563132
	CycC14R	Rhizosphere soil			OP557992	OP629932	OP563133
	CycC15R				OP557993	OP629933	OP563134
*P. multivora*	CycC2		H ^a^		OP557994	OP629934	OP563135
	CycC3				OP557995	OP629935	OP563136
	CycC4				OP557996	OP629936	OP563137
*P. nicotianae*	CycC11R		A2		OP557997	OP629937	OP563138
	CycC12R				OP557998	OP629938	OP563139
	CycC13R		A1		OP557999	OP629939	OP563140

^a^ Homothallic species.

## Data Availability

Data will be available upon specific request to the authors.

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
