# Peer review of "Phytophthora pseudocryptogea, P. nicotianae and P. multivora Associated to Cycas revoluta: First Report Worldwide"

_plants, 2023, doi:10.3390/plants12051197_

Round 1
Reviewer 1 Report
I really enjoyed reading this well written paper, describing an equally well conducted experiment.
I have two important comments
1- Given that all of the research was performe din Sicily, the title absolutely needs to be changed to something like: "Phytophthora stem and root rot, a new disease of Cycas revoluta is first reported from Italy"
2- Please improve figure 4
Otherwise , congrats on anice paper!!!
Reviewer 2 Report
The manuscript by Aloi and colleagues titled “Phytophthora stem and root rot, a new disease of Cycas revoluta worldwide” investigated the cause of disease in C. revoluta by Phytophthora pseudocryptogea. From rotten stems/roots and rhizosphere soil of three- years old pot grown C. revoluta showing dieback symptoms, various techniques were used to bait and to isolate Phytophthora isolates using selective growth media. In addition, DNA barcoding analysis and morphology were used to identify Phytophthora isolates to species level. Three Phytophthora species were identified and by Koch’s postulates demonstrated that P. pseudocryptogea was the causal agent of the decline in the sago palms in Sicily. While P. multivora and P. nicotianae were found in the rhizosphere of symptomatic plants.
Here are some suggestions:
For the title: is the data mainly focused from a nursery based in Sicily and the potential impact to sago palm nurseries worldwide? The title could be modified.
For the discussion:
Apart from isolating P. multivora and P. nicotianae which are known to cause disease in many ornamental plants, it is interesting that P. multivora was not virulent on C. revoluta.
These two recent papers suggests that they are a major problem.
a. Migliorini et al 2019 https://doi.org/10.1016/j.ufug.2019.126460
b. Tsykun, T., Prospero, S., Schoebel, C.N. et al. Global invasion history of the emerging plant pathogen Phytophthora multivora. BMC Genomics 23, 153 (2022). https://doi.org/10.1186/s12864-022-08363-5
Could the authors comment if the P. multivora isolate from Sicily may be different to other isolates found around the world? is this isolate pathogenic to plants found in regions mentioned in references specifically in Sicily? And how could P. multivora been introduced into the nursery?
Baysal-Gurel, F., Bika, R., Simmons, T., & Avin, F. (2022).https://doi.org/10.1094/PDIS-06-21-1342-RE and Kurbetli, İ., Woodward, S., AydoÄŸdu, M., Sülü, G., & Özben, S. (2022). https://doi.org/10.1111/efp.12782 demonstrate how P. nicotianae and P. pseudocryptogea cause problems in nurseries and in forests respectively. In addition, Ferguson, A. J., & Jeffers, S. N. (1999). https://doi.org/10.1094/PDIS.1999.83.12.1129 and Seddaiu S, Brandano A, Ruiu PA, Sechi C, Scanu B. (2020) Forests. 11(9):971. https://doi.org/10.3390/f11090971 demonstrate that container mixers for ornamental nurseries and soil/watercourses in cork oak forests respectively. Could the authors comment how could this be linked to PRR management of nurseries?
For figure 4 page 7, for the Symptom severity graph could the key i.e., Control/CycC2/Cyc11R/CIPH2 be placed at the base of the graph and the image linked to the last data point (a) moved above to see clearly the graph. Or another suggestion is to put the images linked to the data as a supplementary file.
